# *Citrus limon* Peel Powder Reduces Intestinal Barrier Defects and Inflammation in a Colitic Murine Experimental Model

**DOI:** 10.3390/foods10020240

**Published:** 2021-01-25

**Authors:** Nguyen Thi Thanh Tinh, Gertrude Cynthia Sitolo, Yoshinari Yamamoto, Takuya Suzuki

**Affiliations:** 1Graduate School of Biosphere Science, Hiroshima University, 1-4-4 Kagamiyama, Higashi-Hiroshima 739-8528, Japan; tinhntt@dlu.edu.vn (N.T.T.T.); gcsitolo@yahoo.co.uk (G.C.S.); 2Faculty of Agriculture and Forestry, Dalat University, 1 Phu Dong Thien Vuong, Dalat, Lam Dong 670000, Vietnam; 3Department of Physics and Biochemical Sciences, University of Malawi, The Polytechnic, P/Bag 303 Chichiri, Blantyre 3, Malawi; 4Graduate School of Integrated Sciences for Life, Hiroshima University, 1-4-4 Kagamiyama, Higashi-Hiroshima 739-8528, Japan; yamamo59@hiroshima-u.ac.jp

**Keywords:** colitis, dietary fiber, lemon peel, inflammatory cytokine, tight junction

## Abstract

This study examines the ameliorative effects of lemon (*Citrus limon*) peel (LP) powder on intestinal inflammation and barrier defects in dextran sulfate sodium (DSS)-induced colitic mice. The whole LP powder was fractionated into methanol (MetOH) extract and its extraction residue (MetOH residue), which were rich in polyphenolic compounds and dietary fibers, respectively. Mice were fed diets containing whole LP powder, MetOH extract, and MetOH residue for 16 d. DSS administration for 9 d induced bodyweight loss, reduced colon length, reduced the colonic expression of tight junction proteins including zonula occludens-1 and -2, and claudin-3 and -7, and upregulated colonic mRNA expression of interleukin 6, chemokine (C-X-C motif) ligand 2, and C-C motif chemokine ligand 2. Feeding LP powder restored these abnormalities, and the MetOH residue, but not MetOH extract, also showed similar restorations. Feeding LP powder and MetOH residue increased fecal concentrations of acetate and n-butyrate. Taken together, LP powder reduced intestinal damage through the protection of tight junction barriers and suppressed an inflammatory reaction in colitic mice. These results suggest that acetate and n-butyrate produced from the microbial metabolism of dietary fibers in LP powder contributed to reducing colitis.

## 1. Introduction

Inflammatory bowel diseases (IBDs) mainly consist of ulcerative colitis and Crohn’s disease, which are chronic idiopathic disorders causing intestinal inflammation [1]. Patients with IBDs suffer from diarrhea, rectal bleeding, abdominal pain and fever, and experience altering periods of remission and relapse [1]. Originally, the incidence of IBDs was higher in western countries, such as North America and western Europe; however, it is increasing in South America, eastern Europe, Asia, and Africa [2]. These data suggest that both genetic and environmental factors, such as dietary habits, are involved in IBD pathogenesis.

Although IBD pathogenesis is not fully understood, data from basic and clinical studies show that an impaired intestinal barrier triggers intestinal inflammation [3]. An impaired intestinal barrier allows noxious molecules, such as bacterial toxins and dietary antigens, to permeate intestinal tissues, resulting in the robust and chronic activation of immune cells [3]. The intestinal barrier is organized by different barrier components and structures, but the tight junction (TJ) structure expressed in intestinal epithelial cells is one of the major determinants of the intestinal barrier [4]. The TJ structure is a multiple protein complex, consisting of transmembrane and cytosolic plaque proteins, including the transmembrane proteins, occludin, and claudins, whose extracellular loops directly interact with adjacent cells to create a barrier against luminal noxious molecules [5,6]. The intracellular region of the transmembrane protein is bound to cytosolic plaque proteins, such as zonula occludens (ZO), which anchors the TJ complex to the actin cytoskeleton [7]. Although the TJ barrier is regulated by endogenous factors, such as growth factors, cytokines, and hormones [8,9,10], dietary factors, such as polyphenolic compounds and dietary fibers, also have a role in its regulation [11,12,13]. Thus, plant-derived food materials rich in polyphenols and fibers could be developed as a novel tool against intestinal damage and inflammation.

Lemon (*Citrus limon*) is one of the most important fruits in crop production and is widely used as beverages, ice creams, dessert, and cook materials, due to its tart flavor. Lemon fruits contain many bioactive compounds such as vitamin C, citric acid, phenolic compounds, dietary fibers, and essential oil. These compounds often provide us health benefits, including anti-microbial, anti-inflammatory, and anti-oxidant activities [14,15,16]. Thus, lemon fruits have a strong commercial value in the fresh product market and food industry. However, except for edible flesh used in the processing industry, other inedible parts, such as peels, seeds, and pulp matrixes, are mostly wasted as by-products. Previous studies demonstrate that the lemon peel (LP) includes more polyphenolic compounds and dietary fibers than those of the flesh [17,18]. Structurally, the LP consists of the outer layer, called flavedo, and the inner layer, called albedo; the flavedo contains high amounts of polyphenolic compounds, such as hesperidin, diosmin, eriocitrin, and narirutin [18], whereas the albedo is rich in fibers, particularly pectin. Hesperidin, one of the principal polyphenols in lemon, have been shown to possess various biological properties, such as anti-oxidant, anti-inflammatory, and anti-cancer effects [19,20,21]. Diosmin, eriocitrin, and narirutin are also reported to contribute to health care [22,23,24]. In addition, the supplemental dietary fibers influence the activity and composition of intestinal microflora, which are closely related to our health. Our previous studies demonstrated that fermentable fibers reduced the disease symptoms in murine models of colitis and chronic kidney disease [25,26]. Although different mechanisms may exist in the dietary fiber-mediated effects, organic acids produced from the microbial metabolism of dietary fibers regulate the intestinal inflammation and barrier. Thus, LP could be an excellent source of functional food materials to promote human health, however, the roles of LP in the regulation of intestinal homeostasis have not been investigated so far.

The objective of the present study was to examine the ameliorative effect of LP powder on intestinal inflammation and barrier defects in a colitis murine model. In addition, LP powder was separated into two fractions, which were rich in polyphenolic compounds and dietary fibers, respectively, and which fraction had a role in the ameliorative effect on colitis was examined.

## 2. Materials and Methods

### 2.1. Chemicals

Dextran sulfate sodium (DSS; molecular weight: 36,000–50,000) was purchased from MP Biomedicals (Santa Ana, CA, USA). Rabbit anti-ZO-1 (61-7300), occludin (71-1500), claudin-3 (34-1700), claudin-4 (36-4800), claudin-7 (34-9100), and goat Alexa Fluor 488-conjugated anti-rabbit IgG antibodies (A11034) were purchased from Thermo Fisher Scientific (Waltham, MA, USA). Rabbit anti-ZO-2 antibody was purchased from Santa Cruz Biotechnology (sc-11448, Dallas, TX, USA). Horseradish peroxidase-conjugated anti-rabbit IgG antibody was purchased from SeraCare (074-1506, Milford, MA, USA). All other chemicals were obtained from FUJIFILM Wako Pure Chemical Corporation (Osaka, Japan).

### 2.2. Preparation of Citrus Limon Peel Powder and Fractions

Fresh *Citrus limon* peel was purchased from Pokka Sapporo Food and Beverage (Tokyo, Japan). Diced *Citrus limon* peel was freeze-dried and milled to a fine powder with a mill mixer (IFM-66ODG; Iwatani, Osaka, Japan). The resultant powder was passed through a mesh sieve (φ1 mm) and designated as the whole LP powder.

Whole LP powder contained polyphenols and dietary fibers as major bioactive components at high concentrations. To examine which component had a role in the ameliorative effect on colitis, whole LP powder was separated into two fractions rich in polyphenols and fibers, respectively, using a methanol (MetOH) extraction method. Whole LP powder (45 g) was vigorously mixed with 300 mL MetOH, sonicated using an ultrasonic bath (Bransonic 1210; Emerson Electric, St. Louis, MO, USA) for 60 min, and allowed to stand for 16 h. The mixture was centrifuged at 10,000× *g* at 15 °C for 20 min to allow the separation into supernatant and pellet. The pellet was immersed in 300 mL MetOH, vigorously mixed, and centrifuged again. Two additional extractions were performed in the same manner. The pellet obtained through the repeated extractions was air-dried, milled to a fine powder, and designated as the MetOH extraction residue (MetOH residue) of LP powder. The supernatant obtained through the repeated extractions was filtered and evaporated under vacuum at 50 °C. The resultant concentrate was dispersed in 45 g of maize starch powder for faithful handling, air-dried, and designated as MetOH extract of LP powder.

### 2.3. Nutritional Analyses of Citrus limon Peel Powder and the Two Fractions

The nutritional components, including protein, lipid, ash, and moisture in the whole LP powder and the MetOH residue, were analyzed using the Kjeldahl [27], Soxhlet extraction [28], dry ashing (Association of Official Agricultural Chemists [AOAC] 942.05) [29], and air-oven drying methods (AOAC 930.15) [30], respectively. The fiber content in whole LP powder and MetOH residue was determined using the enzymatic-gravimetric method (AOAC 985.29 and 991.43) [31]. Because LP powder is rich in polyphenols, such as hesperidin, eriocitrin, diosmin, and narirutin [17,18], the polyphenols in whole LP powder, MetOH extract, and MetOH residue were determined using liquid chromatography-tandem mass spectrometry (LC/MS/MS) analysis (Acquity UPLC-TQD; Waters, Milford, MA, USA). The UPLC system was fitted with a 1.8-μm C18 column (ACQUITY UPLC BEH C18, 2.1 × 100 mm, Waters) set at 40 °C and a flow rate of 0.3 mL/min. The separation was achieved by gradient elution with methanol-water (30:70) and 100% methanol containing 0.1% formic acid as mobile phases. The auto sampler was kept at 4 °C, and the sample injection volume was 5 μL. The MS analysis was operated by multiple reaction monitoring (MRM) in positive-ion mode. The following MRM transitions were monitored: Hesperidin 611.09 → 303.06, eriocitrin 597.21 → 289.03, diosmin 609.08 → 301.04, and narirutin 581.08 → 273.03. A source temperature of 150 °C and a desolvation temperature of 450 °C were used. The desolvation and cone gas flow were 650 and 35 L/h, respectively.

### 2.4. Animals and Diets

All study protocols were preapproved by the Animal Use Committee of Hiroshima University (authorization No. C19-19), and animal experiments were performed in accordance with Hiroshima University guidelines for the care and use of laboratory animals. Male 7-week old Balb/c mice were purchased from Charles River Japan (Yokohama, Japan). All mice were housed under conditions with a controlled room temperature (20–24 °C), relative humidity (40–60%), and lighting from 8:00 to 20:00 throughout the experimental period. Mice had free access to the AIN-93G-based control diet (Table 1) [32] and distilled water for 1 week during the acclimatization period, prior to the start of the experiment.

Whole LP powder, MetOH extract, and MetOH residue were added to the AIN-93G-based control diet at 5, 6.5, and 4.4% by weight, respectively. LC/MS/MS analysis demonstrated that the recovery ratio of hesperidin, eriocitrin, and diosmin, major polyphenols in the whole LP powder, in the MetOH extract was approximately 97, 77, and 89%, respectively, of those from the original whole LP powder (Table 4). To ensure that these three polyphenols were present at concentrations at least equivalent to those in the 5% whole LP powder diet, the MetOH extract was added to the control diet at 6.5%. Additionally, the fiber content in the MetOH residue increased to 1.13-fold that in the original powder; thus, to ensure that the concentration was approximately equivalent to that in the 5% whole LP powder diet, the MetOH residue was added to the control diet at 4.4%.

### 2.5. Experimental Design

Mice (*n* = 35) were randomly divided into five groups: The control, DSS, DSS + whole LP, DSS + MetOH extract, and DSS + MetOH residue (seven mice/group). The control and DSS groups were fed the control diet for the 16-d experimental period. The DSS + whole LP, DSS + MetOH extract, and DSS + MetOH residue groups were fed the diets containing 5% whole LP powder, 6.5% MetOH extract, and 4.4% MetOH residue by weight, respectively, through the experimental period. Seven days after the start of feeding mice the experimental diets, four DSS treatment groups were administered 2% (*w*/*v*) DSS solution through their drinking water for 9 d to induce experimental colitis, whereas the control group only received distilled water. After DSS administration for 9 d, mice were euthanized by exsanguination under isoflurane anesthesia. The colon was quickly dissected and colon length was measured. The colon was isolated for histological, immunoblot, quantitative reverse transcription-polymerase chain reaction (qRT-PCR), and immunostaining analyses, as described below.

### 2.6. Colitis Clinical Score

To determine the severity of colitis, mouse bodyweight and colitis clinical score were measured every day after the start of DSS administration [16,33]. Briefly, bodyweight loss, diarrhea, bloody stool, and stool consistency were scored from 0 (unaffected) to 4 (severely affected). The detailed scoring system is shown in Table 2.

### 2.7. Histopathology

Mouse colonic tissues were embedded in optimal cutting temperature compounds (Sakura Finetek Japan, Tokyo, Japan), and frozen tissue sections (8 μm-thickness) were prepared on glass slides using a LEICA CM1850 microtome (LEICA, Wetzlar, Germany). Sections were fixed in 4% paraformaldehyde and stained with Mayer’s hematoxylin and eosin using a standard protocol. Images of colonic sections were acquired by a Leica DMI 6000 B microscope (LEICA).

### 2.8. Immunoblot Analysis

A segment of mouse colonic tissues (20 mg) was homogenized in 600 μL lysis buffer with protease and phosphatase inhibitors (1% [*w*/*v*] sodium dodecyl sulfate, 1% [*v*/*v*] Triton X-100, 1% [*w*/*v*] sodium deoxycholate, and 30 mmol/L 2-amino-2-(hydroxymethyl)-1,3-propanediol trimethylolaminomethane) using a Polytron-type homogenizer (Kinematica AG, Lucerne, Switzerland). The supernatant obtained after centrifugation was subjected to immunoblot analyses of TJ proteins, ZO-1, ZO-2, occludin, claudin-3, claudin-4, and claudin-7 using specific antibodies in combination with horseradish peroxidase-conjugated anti-rabbit IgG antibodies. Blots were developed using enhanced chemiluminescence detection reagents (Perkin Elmer Life Sciences, Waltham, MA, USA).

### 2.9. Immunofluorescence

Frozen colonic sections, prepared as described in Section 2.7, were blocked with 4% skimmed milk, and incubated with anti-claudin-3 antibody for 16 h, followed by Alexa Fluor 488-conjugated anti-rabbit IgG antibody for 1 h. The specimens were preserved in mounting fluid, and the fluorescence was visualized using an LCM700 confocal laser scanning microscope (Carl Zeiss, Oberkochen, Germany).

### 2.10. Quantitative Reverse Transcription-Polymerase Chain Reaction (qRT-PCR)

The colonic expression of cytokines and chemokines, such as interleukin 6 (*Il6*), *Il17A*, chemokine (C-X-C motif) ligand 2 (*Cxcl2*), and C-C motif chemokine ligand 2 (*Ccl2*) was determined by qRT-PCR. Total RNA from mouse colonic tissues was isolated using a NucleoSpin^®^ RNA kit (Macherey-Nagel, Düren, Germany) and reverse-transcribed into cDNA using the ReverTra Ace qPCR RT Master Mix kit (Toyobo, Osaka, Japan), according to the manufacturers’ instructions. The PCR reaction was performed in a StepOne Real-Time PCR System (Thermo Fisher Scientific) with 2× Brilliant III Ultra-Fast SYBR Green QPCR Master Mix (Agilent Technologies, Santa Clara, CA, USA) in accordance with the manufacturers’ protocol. The primer sequences are shown in Table 3. The expression of target genes was calculated by the delta delta Ct method using ribosomal protein L13 (*Rpl13*) as the reference gene.

### 2.11. Fecal Organic Acid Analysis

Fresh fecal samples were collected 4 d after the start of DSS treatment for organic acid analysis [34], with minor modifications. Briefly, fecal samples were diluted and homogenized with a 9× volume of distilled water. The supernatant obtained after centrifugation was deproteinized with 50% acetonitrile. Organic acids were chemically derivatized by 3-nitrophenylhydrazine to their 3-nitrophenylhydrazones. Crotonic acid was used as an internal standard. Derivatives of the organic acids (acetate, propionate, butyrate, valerate, lactate, and succinate) were determined by ultraperformance liquid chromatography-mass spectrometry (Waters, Germany).

### 2.12. Statistical Analysis

All data are expressed as mean with a standard error of mean. Statistical analyses were performed using Predictive Analytics Software (PASW) Statistics 18. Statistical differences among groups were determined by one-way analysis of variance (ANOVA) followed by a Tukey-Kramer test. Differences were considered significant at a *p* < 0.05.

## 3. Results

### 3.1. Nutritional Characterization of Whole LP Powder and Its Fractions

The nutritional composition of whole LP powder is shown in Table 4. The sum of protein, lipid, ash, and moisture was 15.9% by weight in whole LP powder. The amount of total dietary fiber in whole LP powder was 47.1%, which included water-soluble fiber at 12.3% and water-insoluble fiber at 34.8%. Protein, lipid, and ash were mostly retained in the MetOH residue of LP powder, and the values were roughly similar to those in whole LP powder. The total dietary fiber amount in the MetOH residue was 53.1%, which was 1.13 times higher than that of the original whole LP powder. This increment might be due to eliminating mono- and disaccharides by MetOH extraction. Protein, ash, and fiber were not analyzed in the MetOH extract, because these nutrients were not extracted by MetOH. Hesperidin, eriocitrin, diosmin, and narirutin in the whole LP powder and MetOH extract, but not MetOH extraction residue, were successfully detected using LC/MS/MS analysis, and the chromatograms were shown in Figure 1. The polyphenol compositions determined in whole LP powder seemed to be consistent with the previous results [17,18], although the compositions varied among the cultivars of lemon. The hesperidin amount in the MetOH extract (423 mg/100 g) was equivalent to that in the whole LP powder (427 mg/100 g), but the amounts of eriocitrin, diosmin, and narirutin were approximately 77, 89, and 51%, respectively, of those in whole LP powder.

### 3.2. Effects of Whole LP Powder and Its Fractions on Bodyweight Loss and Colitis Clinical Score

Experimental colitis was induced by DSS administration 7 d after the start of feeding mice the experimental diets. DSS administration caused bodyweight loss and increased clinical score, indicating the induction of colitis (Figure 2). Bodyweight gain in the DSS group was lower than those in the control group at, and after, day 5. Feeding mice whole LP powder and MetOH residue attenuated bodyweight loss at, and after, day 6. The clinical score in the DSS group was higher than that in the control group at, and after, day 3, and the increments were partially attenuated by feeding mice whole LP powder and MetOH residue. The clinical score in the MetOH extract group was lower than that in the DSS group on days 3 and 4; however, the ameliorative effects on bodyweight gain was not observed.

Colon length is inversely associated with the severity of DSS-induced colitis and is often evaluated as another indicator of colitis. DSS administration shortened colon length in mice; however, feeding mice whole LP powder and MetOH residue attenuated this colon shortening (Figure 3a,b). Based on hematoxylin and eosin staining of colonic specimens, DSS administration caused the loss and distortion of crypts, loss of goblet cells, severe epithelial injury, and inflammatory cell infiltration in the mucosa and submucosa (Figure 3c). Feeding mice whole LP powder and MetOH residue partially, but clearly, attenuated these abnormal symptoms, and the crypt structures were relatively preserved in these two groups. Although the MetOH extract also reduced DSS-induced histological changes, the reduction looked weaker than those in the whole LP powder and the MetOH residue groups.

### 3.3. Effect of Whole LP Powder and Its Fractions on the Colonic TJ Barrier

An impaired intestinal TJ barrier has a role in the pathogenesis of IBDs [3]. The colonic expression of ZO-1, ZO-2, claudin-3, and claudin-7 in the DSS group was lower than those in the control group (Figure 4). DSS administration reduced occludin expression, but the reduction was not statistically significant. Feeding mice whole LP powder restored the decreased expression of these TJ proteins, although the restoration in claudin-7 expression was not significant. Feeding mice MetOH residue also restored ZO-2, claudin-3, and claudin-7 expression. Meanwhile, any TJ proteins examined in the DSS + MetOH extract group did not differ from those in the DSS group. Because claudin-3 is one of the major isoforms expressed in the colonic epithelium and whole LP powder and MetOH residue significantly restored its expression, its cellular localization was examined by immunofluorescence analysis (Figure 5). In the control group, claudin-3 was observed at basolateral, as well as in the junctional regions of epithelial cells. DSS administration impaired the expression and localization, while feeding mice whole LP powder and MetOH residue, but not the MetOH extract, partially, but clearly, restored them. These data suggest that the restoration of the colonic TJ barrier by whole LP powder and MetOH residue was involved in reducing colitis.

### 3.4. Effect of Whole LP Powder and Its Fractions on Colonic Gene Expression

An impaired colonic TJ barrier allows luminal noxious molecules to permeate into colonic mucosal tissues and causes robust and uncontrolled expression of inflammatory cytokines and chemokines [35]. DSS administration upregulated expression of *Il6*, *Il17A*, *Cxcl2*, and *Ccl2* in colonic tissues, whereas feeding mice whole LP powder and MetOH residue suppressed expression of *Il6*, *Cxcl2*, and *Ccl2*, but not *Il17A* (Figure 6). These results suggest that the suppression of the inflammatory reaction was another mechanism underlying the reduction of colitis. The MetOH extract modestly suppressed *Cxcl2* expression, but did not influence the other genes.

### 3.5. Effect of Whole LP Powder and Its Fractions on Fecal Organic Acids

Organic acids (acetate, propionate, butyrate, valerate, lactate, and succinate) are major metabolites of undigested carbohydrates from intestinal microorganisms [34]. In particular, acetate, propionate, butyrate, and valerate are categorized into short-chain fatty acids (SCFAs). Because the MetOH residue, rich in dietary fibers, exhibited ameliorative effects in colitic mice in a manner partially similar to that of whole LP powder, we speculated that the microbial metabolism of dietary fibers in whole LP powder was involved in reducing colitis. Acetate, n-butyrate, iso-butyrate, and lactate in the DSS + whole LP group and acetate, n-butyrate, and lactate in the DSS + MetOH residue group were higher than those in the DSS group (Figure 7). Succinate in the DSS + whole LP and DSS + MetOH residue groups were or tended to be lower than that in the DSS group. There was no difference in propionate, n-valerate, and iso-valerate among the groups.

## 4. Discussion

IBDs affect the health of millions of people and pose a considerable health burden worldwide [2]. Currently, aminosalicylates, glucocorticoids, immunosuppressive agents, and biological drugs are used to treat IBDs; however, these treatments often involve severe side-effects, an insufficient response, and high cost [36]. Therefore, the development of novel preventive or therapeutic approaches using natural herbal medicines and dietary supplements is desired. This study demonstrated that whole LP powder, rich in polyphenolic compounds and dietary fibers, attenuated an intestinal barrier defect and inflammation in a murine model of colitis. The LP-mediated ameliorative effects seemed to be involved in the protection of the colonic TJ barrier and suppression of an inflammatory reaction. In addition, our results indicated that dietary fibers in whole LP powder had an important role in reducing colitis. In the previous studies, the peel extracts of other citrus fruits, such as orange (*Citrus sinensis*), yuzu (*Citrus junos Tanaka*), and Kawachibankan (*Citrus kawachiensis*), also reduced the intestinal inflammation in colitic mice [16,37,38]. In agreement with our findings, it is suggested that dietary fibers in these peels are at least in part involved in the regulation of intestinal inflammation.

One of the most important findings of the present study was that the MetOH residue, rich in dietary fibers, exhibited ameliorative effects on colitis in a manner similar to that of whole LP powder. In addition, feeding mice the MetOH residue, as well as whole LP powder, influenced some organic acids in feces. In particular, acetate, n-butyrate, and lactate were increased, and succinate was decreased by both whole peel powder and MetOH residue. These data suggest that the alterations of these organic acids, at least in part, contributed to the whole LP powder-mediated reduction of colitis. Our previous study demonstrates that feeding mice fermentable fibers leading to increased luminal SCFAs reduces intestinal inflammation and barrier defects in colitic mice [25]. Clinical studies have also shown lower fecal SCFA levels in patients with IBDs than those in healthy subjects [39,40].

At least two mechanisms are involved in the organic acid-mediated reduction of colitis: The regulation of the colonic TJ barrier and inflammatory cytokine production. The intestinal TJ barrier limits the permeation of luminal noxious molecules into mucosal tissues and prevents uncontrolled activation of immune cells [35]. Feeding mice whole LP powder and MetOH residue restored the structure of the colonic TJ barrier. Organic acids have been reported to regulate the integrity and structure of the intestinal TJ barrier [41,42,43,44]. Acetate, n-butyrate, and lactate enhance TJ barrier integrity in rat cecum and intestinal Caco-2 cells [41]. Whereas, succinate, which was increased by DSS administration, have been shown to induce the hyperpermeability of intestinal TJ [41]. Additionally, feeding mice whole LP powder and MetOH residue suppressed inflammatory cytokine expression, including *Il6* and *Cxcl2* (a murine homolog of *Il8*). SCFAs, including acetate and n-butyrate, but not lactate, have been reported to suppress the inflammatory response of intestinal epithelial cells [45]. Stimulation of intestinal Caco-2 cells and mouse colonic tissues with tumor necrosis factor-α induces inflammatory cytokine production, including IL-6 and IL-8, but treatment with acetate and n-butyrate reduces cytokine production [45]. IL-8, secreted by epithelial cells, recruits neutrophils into inflamed colonic tissues, where neutrophils develop mucosal inflammation through intercellular interactions with other cells, such as macrophages and T cells [46]. In clinical studies, various immune cells, including neutrophils, accumulate in inflamed tissues of patients with ulcerative colitis [47]. Administration of a neutralizing antibody against the IL-8 receptor and C-X-C motif chemokine receptor 2 attenuates disease symptoms in DSS-induced colitic mice [48]. Although the DSS administration and whole LP powder modestly influenced the fecal iso-butyrate, its physiological role is poorly understood. Thus, the acetate and n-butyrate-mediated reduction of inflammatory cytokines, such as IL-6 and IL-8, seems to be another mechanism underlying the reduction of colitis.

Accumulating evidence indicates that the intestinal microflora has an important role in the pathogenesis of IBDs [49,50]. The present study did not examine the effects of LP powder on the intestinal microflora, but the alteration of fecal SCFAs implied that the supplemental LP possibly influenced the activity or composition of intestinal microflora. Some studies show the distinct composition of intestinal microflora in patients with IBDs from healthy people [49]. The patients with ulcerative colitis often show the reduced abundance of “protective bacteria”, such as lactobacilli and bifidobacteria [49,50]. Oral administration of these bacteria reduces the disease symptoms both in the animal models of colitis and patients with IBDs [51,52,53]. On the other hand, sulfate-reducing bacteria have been detected more frequently in the patients with IBDs, compared to healthy people [54]. The involvement of sulfate-reducing bacteria in intestinal inflammation remains unclear, but it seems that the hydrogen sulfide produced by these bacteria induces mucosal damage [55]. To understand the precise roles of LP powder in the regulation of intestinal inflammation and barrier, the intestinal microflora in mice fed LP powder should be analyzed in future studies.

Although we hypothesized that dietary fiber has an important role in whole LP powder-mediated reduction of colitis, other bioactive components, such as polyphenols, also marginally contributed. This was because the restoration of TJ proteins, such as ZO-1 and occludin by the MetOH residue, was not as effective as that by whole LP powder, even though both the diets included the same levels of dietary fibers. In addition, feeding mice MetOH extract partly, but significantly, reduced DSS-induced *Cxcl2* expression in colonic tissues. Previous studies showed that oral administration of hesperidin (10–40 mg/kg bodyweight) and eriocitrin (20 mg/kg bodyweight) reduces disease symptoms in DSS-induced colitic mice [56,57]. The doses of hesperidin and eriocitrin achieved through feeding mice whole LP powder and MetOH extract were approximately equivalent to 17 and 7 mg/kg bodyweight, respectively, under the feeding conditions of this study. At least, hesperidin in whole LP powder and MetOH extract could be expected to exhibit ameliorative effects on colitic mice [57]. The reason for the weaker ameliorative effect of the MetOH extract than that was expected might be due to minor differences in experimental conditions. Previously, hesperidin was administered to C57BL/6 mice by oral gavage [57], whereas, in this study, whole LP powder was mixed in the diet and provided to Balb/c mice.

Our results suggest that SCFAs target the TJ barrier and inflammatory reaction in the colonic epithelium and mediate the protective effect of whole LP powder in colitic mice; however, additional mechanisms may also exist. N-butyrate promotes the expansion of regulatory T cells, which suppress an inflammatory reaction [58]. Acetate also induces the intestinal production of immunoglobulin A, which protects intestinal tissue from inflammation [59]. Possible involvement of these mechanisms should be investigated in further studies.

## 5. Conclusions

This study examined the ameliorative effects of LP powder on intestinal inflammation and barrier defects, and the bioactive components responsible for the effect were examined in DSS-induced colitic mice. Results showed that the supplemental feeding of LP powder reduced the intestinal inflammation and damage in colitic mice. Although the LP powder is rich in both polyphenols and dietary fibers, the dietary fibers in LP powder largely contributed to reducing colitis. Although the underlying mechanisms were not fully understood, our results suggest that SCFAs produced from colonic fermentation of dietary fibers in LP powder protected the colonic TJ barrier and suppressed an inflammatory response. Further studies will investigate the possible application of LP powder to treat intestinal damage and inflammation.

## Figures and Tables

**Figure 1 foods-10-00240-f001:**
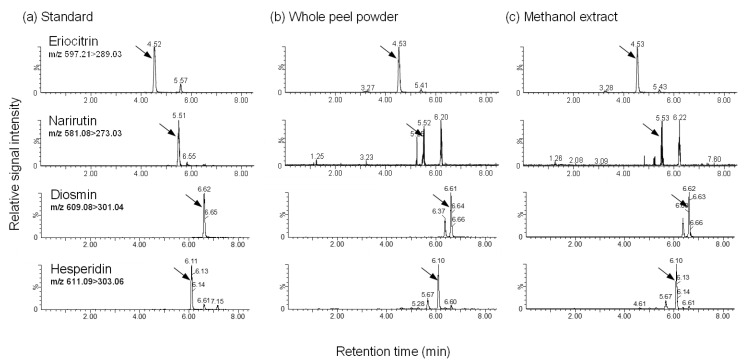
Representative chromatograms of polyphenols in standard solution, whole peel powder, and its methanol extract from UPLC/MS/MS analysis. The standard solution (50 μM, **a**), whole peel powder (**b**), and its methanol extract (**c**) were analyzed. Arrows indicate the peaks of each polyphenol.

**Figure 2 foods-10-00240-f002:**
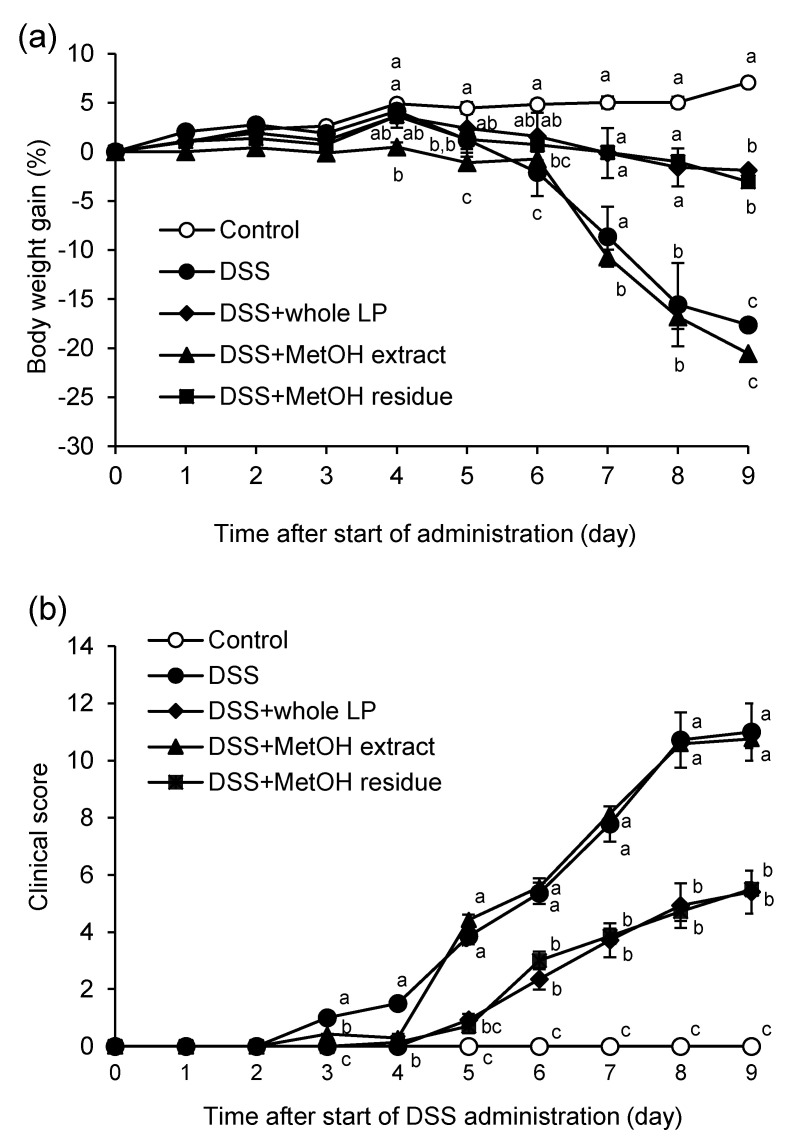
Effects of whole LP powder, methanol (MetOH) extract, and MetOH extraction residue (MetOH residue) on bodyweight change and clinical score in dextran sodium sulfate (DSS)-induced colitic mice. Bodyweight change (**a**) and clinical score (**b**) of mice fed diets with and without whole LP powder, MetOH extract, and MetOH residue, with or without DSS administration. Values are the mean ± SEM (*n* = 7). Means without a common letter differ, *p* < 0.05.

**Figure 3 foods-10-00240-f003:**
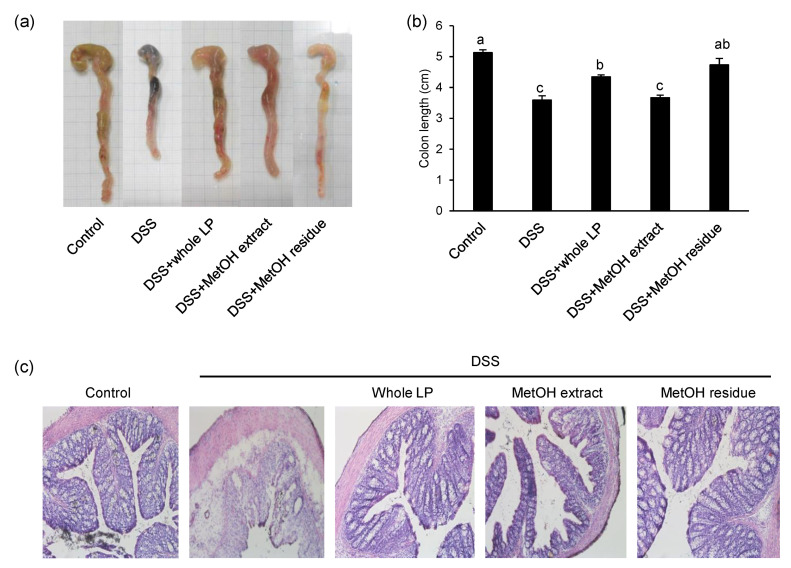
Effects of whole LP powder, methanol (MetOH) extract, and MetOH extraction residue (MetOH residue) on colon length and mucosal structure in dextran sodium sulfate (DSS)-induced colitic mice. Colon length (**a**,**b**) of mice fed diets with and without whole LP powder, MetOH extract, and MetOH residue, with or without DSS administration. Representative images of the cecum and colon of seven mice in each group are shown in (**a**). Scale bar represents 10 mm. Colonic sections of mice were stained with hematoxylin and eosin. Representative images of seven mice in each group are shown in (**c**). Scale bar represents 100 μm. Values are the mean ± SEM (*n* = 7). Means without a common letter differ, *p* < 0.05.

**Figure 4 foods-10-00240-f004:**
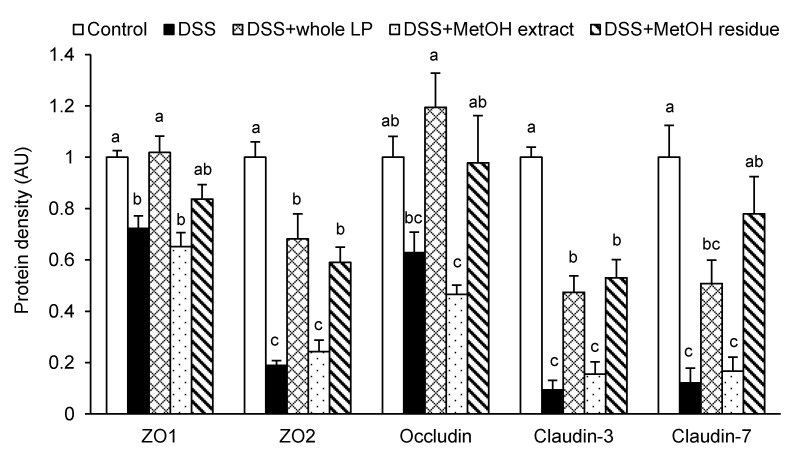
Effects of whole LP powder, methanol (MetOH) extract, and MetOH extraction residue (MetOH residue) on tight junction protein expression in the colon of dextran sodium sulfate (DSS)-induced colitic mice. Protein expression of zonula occludens (ZO)-1, ZO-2, occludin, claudin-3, and claudin-7 in the colon of mice fed diets with and without whole LP powder, MetOH extract, and MetOH residue, with or without DSS administration, as determined by immunoblot analysis. Values are the mean ± SEM (*n* = 7). Means without a common letter differ, *p* < 0.05.

**Figure 5 foods-10-00240-f005:**
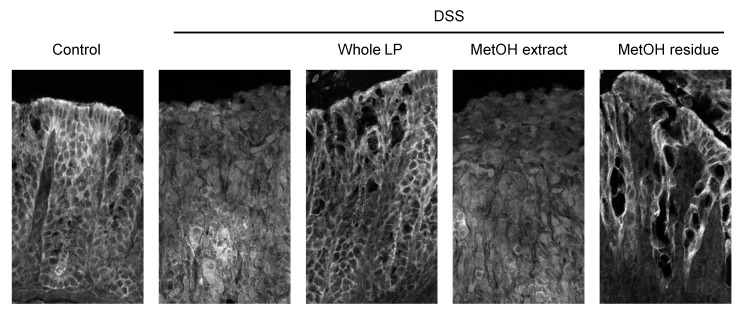
Effects of whole LP powder, methanol (MetOH extract), and MetOH extraction residue (MetOH residue) on claudin-3 expression in the colon of dextran sodium sulfate (DSS)-induced colitic mice. Immunolocalization of claudin-3 in the colon of mice fed diets with and without whole LP powder, MetOH extract, and MetOH residue, with or without DSS administration, as analyzed by immunofluorescence microscopy. Representative images of seven mice in each group are shown.

**Figure 6 foods-10-00240-f006:**
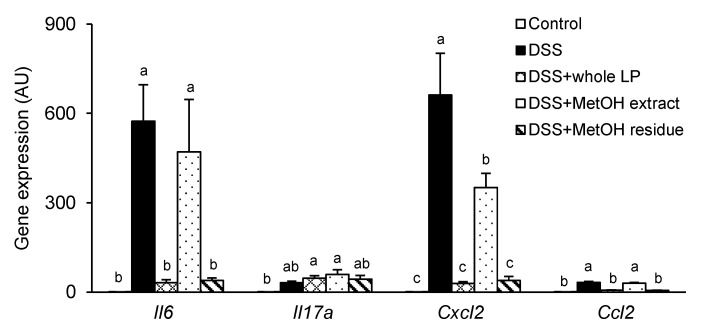
Effects of whole LP powder, methanol (MetOH) extract, and MetOH extraction residue (MetOH residue) on inflammatory cytokine expression in the colon of dextran sodium sulfate (DSS)-induced colitic mice. Gene expression of *Il6*, *Il17A*, *Cxcl2*, and *Ccl2* in the colon of mice fed diets with and without whole LP powder, MetOH extract, and MetOH residue, with or without DSS administration, as determined by real-time quantitative polymerase chain reaction (RT-qPCR) analysis. Values are the mean ± SEM (*n* = 7). Means without a common letter differ, *p* < 0.05.

**Figure 7 foods-10-00240-f007:**
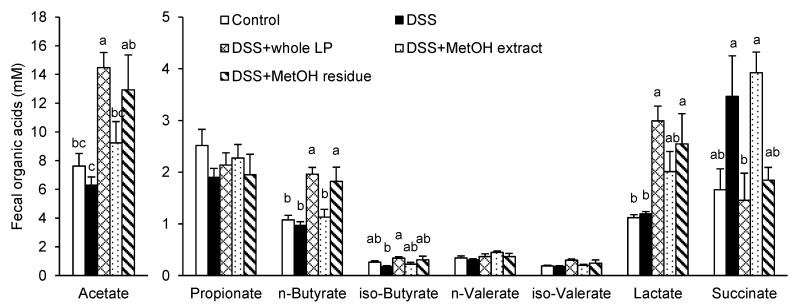
Effects of whole LP powder, methanol (MetOH) extract, and MetOH extraction residue (MetOH residue) on organic acid concentrations in feces of dextran sodium sulfate (DSS)-induced colitic mice. Fecal concentrations of acetate, propionate, n-butyrate, iso-butyrate, n-valerate, iso-valerate, lactate, and succinate in mice fed diets with and without whole LP powder, MetOH extract, and MetOH residue, with or without DSS administration, as determined by liquid chromatograph/tandem mass spectrometry (LC/MS/MS) analysis. Values are the mean ± SEM (*n* = 7). Means without a common letter differ, *p* < 0.05.

**Table 1 foods-10-00240-t001:** Composition of experimental diets ^a^.

Ingredient	Control Diet
	g/kg diet
Casein ^b^	200
α-Corn starch ^c^	529.5
Sucrose	100
Soybean oil	70
Choline bitartrate	2.5
l-Cystine	3
Mineral mixture ^d^	35
Vitamin mixture ^d^	10
Cellulose ^e^	50

^a^ Whole LP powder, the MetOH extract, and the MetOH extraction residue were added to the control diet at 5, 6.5, and 4.4%, respectively, by substitution for equal amounts of starch. ^b^ Casein (ALACID; New Zealand Daily Board, Wellington, New Zealand). ^c^ α-Corn starch (Amylalpha CL; Chuo-Shokuryou Co. Ltd., Aichi, Japan). ^d^ Mineral and vitamin mixtures were prepared according to the AIN-93G formulation. ^e^ Powdered cellulose (Just fiber; International Fiber Corporation, New York, NY, USA).

**Table 2 foods-10-00240-t002:** Clinical scoring system.

Score	Diarrhea Stool	Bloody Stool	Weight Loss (% of Initial)
0	Normal	Normal color	<1
1	Mildly soft	Brown color	1–5
2	Very soft	Reddish color	6–10
3	Watery	Bloody stool	11–20
4	More watery	More bloody	>21

The sum of the scores of three parameters was defined as the clinical score.

**Table 3 foods-10-00240-t003:** Primer sequences used for qRT-PCR analysis.

Target Gene	Forward	Reverse
Mouse *Il6*	5′-CTGATGCTGGTGACAACCAC-3′	5′-TCCACGATTTCCCAGAGAAC-3′
Mouse *Il17a*	5′-AGCTGGACCACCACTTGAAT-3′	5′-ACACCCACCAGCATCTTCTC-3′
Mouse *Ccl2*	5′-GGAATGGGTCCAGACATACATTA-3′	5′-TAGCTTCAGATTTACGGGTCAAC-3′
Mouse *Cxcl2*	5′-AGTGAACTGCGCTGTCAATG-3′	5′-ACTTTTTGACCGCCCTTGAG-3′
Mouse *Rpl13*	5′-TCAAGAAGGTGGTGAAGCAG-3′	5′-AAGGTGGAAGAGTGGGAGTTG-3′

**Table 4 foods-10-00240-t004:** Nutritional composition of whole lemon peel (LP) powder, the MetOH extract, and the MetOH extraction residue.

	Whole LP Powder	MetOH Extract	MetOH Extraction Residue
Protein (g/100 g powder)	0.7	N.A.	0.7
Lipid (g/100 g powder)	4.3	N.A.	5.8
Ash (g/100 g powder)	3.9	N.A.	4.0
Dietary fiber (g/100 g powder)	47.1	N.A.	53.1
Water soluble	12.3	N.A.	17.8
Water insoluble	34.8	N.A.	35.3
Moisture	7.0	N.A.	9.2
Polyphenols (mg/100 g powder)			
Hesperidin	427	423	N.D.
Eriocitrin	174	134	N.D.
Diosmin	146	130	N.D.
Narirutin	26.5	13.5	N.D.

N.A., not analyzed. N.D., not detected.

## Data Availability

Data available on request.

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
