# Peer review of "Citrus limon Peel Powder Reduces Intestinal Barrier Defects and Inflammation in a Colitic Murine Experimental Model"

_foods, 2021, doi:10.3390/foods10020240_

Round 1

Reviewer 1 Report

Dear Authors,
I have carefully reviewed your manuscript.
The manuscript is interesting and describes citrus limon peel powder reduces intestinal barrier defects and inflammation in a colitic murine experimental model. The authors obtained interesting results.

However, it should be indicate more information about the etiological role of microbial communities in the IBD development in discussions part of the manuscript. Please more discuss the role of sulfate-reducing bacteria and produced hydrogen sulfide in the development of the process. What is the interaction with lactic acid bacteria during ulceractive colitis? Which is pH if large intestine during the colitis?

What is the effect of citrus limon peel powder on microbial communities?

The following references could be useful for authors:
https://pubmed.ncbi.nlm.nih.gov/12691258/
https://pubmed.ncbi.nlm.nih.gov/30707247/
https://pubmed.ncbi.nlm.nih.gov/19709217/
https://pubmed.ncbi.nlm.nih.gov/30191181/
DOI https://doi.org/10.2478/s11756-018-0118-2
https://academic.oup.com/femsle/article/86/2/103/517375
https://pubmed.ncbi.nlm.nih.gov/31330956/
https://pubmed.ncbi.nlm.nih.gov/30775453/
https://academic.oup.com/femsec/article/12/2/117/503522
https://pubmed.ncbi.nlm.nih.gov/32575467/
https://pubmed.ncbi.nlm.nih.gov/31756903/
https://pubmed.ncbi.nlm.nih.gov/32178484/
https://pubmed.ncbi.nlm.nih.gov/31614543/
http://dx.doi.org/10.1016/j.jare.2020.03.003

Please to discus little bit the role microbial communities in IBD development and under the effect of citrus limon peel powder using these references.

Best wishes, the reviewer.

Author Response

Reply to Reviewer1

Thank you for your kind consideration for our manuscript. We are very pleased to receive the opportunity to improve our manuscript. We have carefully read your comments and made the revised version. All revisions are highlighted by the “Track change function” in MS-Word. The specific corrections and answers to your comments are described below.

Comments #1. However, it should be indicate more information about the etiological role of microbial communities in the IBD development in discussions part of the manuscript. Please more discuss the role of sulfate-reducing bacteria and produced hydrogen sulfide in the development of the process. What is the interaction with lactic acid bacteria during ulceractive colitis? Which is pH if large intestine during the colitis?

Thank you for your important comments. As you suggested, we have added one paragraph, in which the roles of intestinal microbiota in the IBD pathogenesis are discussed, in the discussion section of the revised manuscript. Especially, we described the roles of lactobacilli, bifidobacterial, and sulfate-reducing bacteria in IBDs, with citing some previous works. According to your comment, we measured the fecal pH of mice, but we did not find any differences among groups. Therefore, we did not add the pH measurement in the revised version. Revisions about these issues are described below.

L412-424 in the revised manuscript.

Accumulating evidence indicates that the intestinal microflora has an important role in the pathogenesis of IBDs [49,50]. The present study did not examine the effects of LP powder on the intestinal microflora, but the alteration of fecal SCFAs implied that the supplemental LP possibly influenced the activity or composition of intestinal microflora. Some studies show the distinct composition of intestinal microflora in patients with IBDs from the healthy people [49]. The patients with ulcerative colitis often show the reduced abundance of “protective bacteria” such as lactobacilli and bifidobacteria [49,50] . Oral administration of these bacteria reduces the disease symptoms both in the animal models of colitis and patients with IBDs [51-53]. On the other hand, sulfate-reducing bacteria have been detected more frequently in the patients with IBDs in compared to healthy people [54]. The involvement of sulfate-reducing bacteria in the intestinal inflammation remains unclear, but it seems that the hydrogen sulfide produced by these bacteria induces the mucosal damage [55]. To understand the precise roles of LP powder in the regulation of intestinal inflammation and barrier, the intestinal microflora in mice fed LP powder should be analyzed in future studies.

Comment #2. What is the effect of citrus limon peel powder on microbial communities?

Thank you for your question. We also speculated that LP powder influenced the intestinal microflora of mice. However, the microbial analysis by qPCR was not successful, maybe due to the presence of DSS in our fecal samples. I heard that the DSS often interferes the PCR reaction. We are going to find the solution for our trouble and run the NGS to analyze the microflora in the future study. I wish you kindly understand our situation. Thank you again.

Reviewer 2 Report

This is an interesting study whose topic coincides with the scope of the journal.

This research aims to the determination of the effect of a fruit by-product, lemon peels, on the intestinal function, as well as the identification of the bioactive compounds responsible for this effect. This study examined the ameliorative effect of two fractions lemon peel’s powder (a fraction rich in polyphenolic compounds and the other rich in dietary fibers) on intestinal inflammation and barrier defects in a colitis murine model. Authors demonstrate that dietary fibers from lemon peels play an important role on the reduction of intestinal inflammation and damage in colitic mice.

However, there are some remarks about this paper:

1. The abstract is a little disorganized. I suggest the use of more concise and understandable sentences.

2. In the introduction, I liked to see a well-structured paragraph about lemon peels than that written by the authors. Several studies have been also carried out on lemon peels, their chemical composition, their biological and pharmacological properties, as well as their applications in various fields. Also, the objective should be reorganised and must be clearer.

3. In the section 2.3 of Materials et Methods, it is suggested to add some details about the conditions of liquid chromatography tandem mass spectrometry (LC/MS/MS) analysis.

4. What are the reasons for the differences observed in the composition of whole lemon peel (LP) powder, MetOH extract, and MetOH extraction residue? For example, why total dietary fiber amount was higher in the MetOH residue?

5. In the section 3.1 of Results, the chemical composition of lemon peels and extracts should more detailed. Authors can add a comparison with others studies conducted on Citrus peels.

I suggest the addition of the chromatograms of liquid chromatography.

6. The authors can strengthen this paper by adding some comparisons with other studies carried out on the effect of fruit peel on intestinal inflammation and barrier defects in colitis.

7. The conclusion is too short. The authors must highlight the objectives achieved by the interesting results of this study.

Author Response

Dear Reviewer2,

Thank you for your consideration. Please find a PDF file describing our replies to your comments.

Sincerely,

Takuya Suzuki
